# Efficient and robust coding in heterogeneous recurrent networks

**Fleur Zeldenrust** [1]\*, **Boris Gutkin** [2,3], **Sophie Denéve** [2]

**1** Department of Neurophysiology, Donders Institute for Brain, Cognition and Behaviour, Radboud University, Nijmegen, the Netherlands, **2** Group for Neural Theory, INSERM U960, Département d'Études Cognitives, École Normal Supérieure PSL University, Paris, France, **3** Center for Cognition and Decision Making, National Research University Higher School of Economics, Moscow, Russia

\* f.zeldenrust@neurophysiology.nl

**Data Availability Statement:** All code to generate the data can be found at https://github.com/fleurzeldenrust/Efficient-coding-in-a-spiking-predictive-coding-network.

## Abstract

Cortical networks show a large heterogeneity of neuronal properties. However, traditional coding models have focused on homogeneous populations of excitatory and inhibitory neurons. Here, we analytically derive a class of recurrent networks of spiking neurons that close to optimally track a continuously varying input online, based on two assumptions: 1) every spike is decoded linearly and 2) the network aims to reduce the mean-squared error between the input and the estimate. From this we derive a class of predictive coding networks, that unifies encoding and decoding and in which we can investigate the difference between homogeneous networks and heterogeneous networks, in which each neurons represents different features and has different spike-generating properties. We find that in this framework, 'type 1' and 'type 2' neurons arise naturally and networks consisting of a heterogeneous population of different neuron types are both more efficient and more robust against correlated noise. We make two experimental predictions: 1) we predict that integrators show strong correlations with other integrators and resonators are correlated with resonators, whereas the correlations are much weaker between neurons with different coding properties and 2) that 'type 2' neurons are more coherent with the overall network activity than 'type 1' neurons.

## Author summary

Neurons in the brain show a large diversity of properties, yet traditionally neural network models have often used homogeneous populations of neurons. In this study, we investigate the effect of including heterogoenous neural populations on the capacity of networks to represent an input stimulus. We use a predictive coding framework, by deriving a class of recurrent filter networks of spiking neurons that close to optimally track a continuously varying input online. We show that if every neuron represents a different filter, these networks can represent the input stimulus more efficiently than if every neuron represents the same filter.

**Funding:** FZ acknowledges support from the Netherlands Organisation for Scientific Research (Nederlandse Organisatie voor Wetenschappelijk Onderzoek, NWO) Veni grant (863.150.25) and the Radboud University (Christine Mohrmann Foundation), SD acknowledges support from Neuropole Region Île de France (NERF) and ERC consolidator grant "predispike" and BSG acknowledges funding from the Basic Research Program at the National Research University Higher School of Economics (HSE University), ANR-17-EURE- 1553-0017, and ANR-10-IDEX-0001-02. The funders had no role in study design, data collection and analysis, decision to publish, or preparation of the manuscript.

## Introduction

It is widely accepted that neurons do not form a homogeneous population, but that there is large variability between neurons. For instance, the intrinsic biophysical properties of neurons, such as the densities and properties of ionic channels, vary from neuron to neuron [1–5]. Therefore, the way in which individual neurons respond to a stimulus (their 'encoding' properties or receptive field) also varies. A classical example is the difference between 'type 1' and 'type 2' neurons [6–10]: 'type 1' neurons or 'integrators' respond with a low firing frequency to constant stimuli, which they increase with the amplitude of the stimulus. 'Type 2' neurons or 'resonators' on the other hand respond with an almost fixed firing frequency, and are sensitive to stimuli in a limited frequency band. Apart from these intrinsic 'encoding' properties of neurons, the 'decoding' propertes of neurons also show a large variability (see for instance [11]). Even if we do not take various forms of (short term) plasticity into account, there is a large heterogeneity in the shapes of post-synaptic potentials (PSPs) that converge onto a single neuron (for an overview: [12, 13]), depending on amongst others: the projection site (soma/dendrite: [14]), the number of receptors at the synapse, postsynaptic cell membrane properties ([15–17]), the type of neurotransmitter (GABAA, GABAB, glutamate), synapse properties (channel subunits, [18]), the local chloride reversal potential and active properties of dendrites [12]. This heterogeneity results in variability of decay times, amplitudes and overall shapes of PSPs. So neural heterogeneity plays an important role both in encoding and in decoding stimuli. Whereas the study of homogeneous networks has provided us with invaluable insights [19–22], the effects of neural heterogeneity on neural coding have only been studied to a limited extent [23–25]. Here we show that networks with spiking neurons with heterogeneous encoding and decoding properties can do optimal online stimulus representation. In this framework, neural variability is not a problem that needs to be solved, but it increases the networks' versatility of coding.

In order to investigate coding properties of neurons and networks, we need to use a framework in which we can assess the encoding and decoding properties and the network properteis. To characterize the relationship between neural stimuli and responses, filter networks (such as the Linear-Nonlinear Poisson (LNP) model [26], [27] and the Generalized Linear Model (GLM) [28, 29], for an overview, see [30], [31]) are widely used. In these models, each neuron in a network compares the input it receives with an 'input filter'. If the two are similar enough, a spike is fired. In a GLM, unlike the LNP model, this output spike train is filtered and fed back to the neuron, thereby incorporating effectively both the neuron's receptive field and history-dependent effects such as the refractory period and spike-frequency adaptation. It can be shown, that these types of filter-frameworks describe a maximum-likelihood relation between the input and the output spikes [32], [33]. However, these models are purely descriptive: they only describe how spike trains are generated, not how they should be read out or 'decoded'. In this paper, we analytically derive a class of recurrent networks of spiking neurons that close to optimally track a continuously varying input online. We start with two very simple assumptions: 1) every spike is decoded linearly and 2) the network aims to perform optimal stimulus representation (i.e. reduces the mean-squared error between the input and the estimate). From this we derive a class of predictive coding networks, that unifies encoding (how a network represents its input in its output spike train) and decoding (how the input can be reconstructed from the output spike train) properties.

We investigate the difference between homogeneous networks, in which all neurons represent similar features in the input and have similar response properties, and heterogeneous networks, in which neurons represent different features and have different spike-generating properties. Firstly, we assess the properties of single-neuron (spike-triggered average, input-output frequency curve and phase-response curve) and show that in this framework, 'type 1' or

'integrator' [6, 34] neurons and resonators or 'type 2' neurons arise quite naturally. We show that the response properties of these neurons (the encoding properties, [35]) and dynamics of the PSPs they send (the decoding properties) are inherently linked, thereby giving a functional interpretation to these classical neuron types. Next, we investigate the effects of these different types of neurons on the network behaviour: we investigate the coding efficiency, robustness and trial-to-trial variability in *in-vivo*-like simulations. Finally, we predict that 1) integrators show strong correlations with other integrators and resonators are correlated with resonators, whereas the correlations are much weaker between neurons with different coding properties [36] and 2) that 'type 2' neurons are more coherent with the overall network activity than 'type 1' neurons.

## Materials and methods

We analytically derive a recurrent network of spiking neurons that close to optimally tracks a continuously varying input online. We start with two assumptions: 1) every spike is decoded linearly and 2) the network aims to perform optimal stimulus representation (i.e. reduces the mean-squared error between the input and the estimate). We construct a cost function that consists of three terms: 1) the mean-squared error between the stimulus and the estimate, 2) a linear cost that punishes high firing rates and 3) a quadratic cost that promotes distributed firing. Every spike that is fired in the network reduces this cost function. The code for simulating such a network can be found at GitHub: github.com/fleurzeldenrust/Efficient-coding-in-a-spiking-predictive-coding-network.

### Derivation of a filter-network that performs stimulus estimation

Suppose we have a set of $N$ neurons j that use filters $g_j(t)$ to represent their input. Their spiking will give an estimated input equal to

$$\hat{s}(t) = \sum_{j=1}^{N} g_j(t) * \rho_j^{\text{ideal}}(t) = \sum_{j=1}^{N} \sum_{i=1}^{n_j} g_j(t - t_j^i),\tag{1}$$

where $t_j^i$ are the spike times $i$ of neuron $j$ (what 'ideal' stands for will be explained later). The mean-squared error between the estimate $\hat{s}$ and the stimulus $s$ equals

$$E(t) = \int_0^t \left(s(t) - \hat{s}(t)\right)^2 dt = \int_0^t \left(s(t) - \sum_{j=1}^{N} \sum_{i=1}^{n_j} g_j(t - t_j^i)\right)^2 dt.\tag{2}$$

Suppose that at time $t = T + \Delta$ we want to estimate the difference in error given that there was a spike of neuron $m$ at time $T$ or not. The difference in error is given by

$$
\begin{aligned}
&E^{\text{no spike at } T}(T + \Delta) - E^{\text{spike at } T}(T + \Delta) = \\
&\int_0^{T+\Delta} \left(s(t) - \sum_{j=1}^{N} \sum_{i=1}^{n_j} g_j(t - t_j^i)\right)^2 dt \\
&- \int_0^{T+\Delta} \left(s(t) - \sum_{j=1}^{N} \sum_{i=1}^{n_j} g_j(t - t_j^i) - g_m(t - T)\right)^2 dt = \\
&- \int_0^{T+\Delta} g_m(t - T)^2 dt \\
&+ 2 \int_0^{T+\Delta} g_m(t - T)\left(s(t) - \sum_{j=1}^{N} \sum_{i=1}^{n_j} g_j(t - t_j^i)\right) dt.
\end{aligned}
\tag{3}
$$

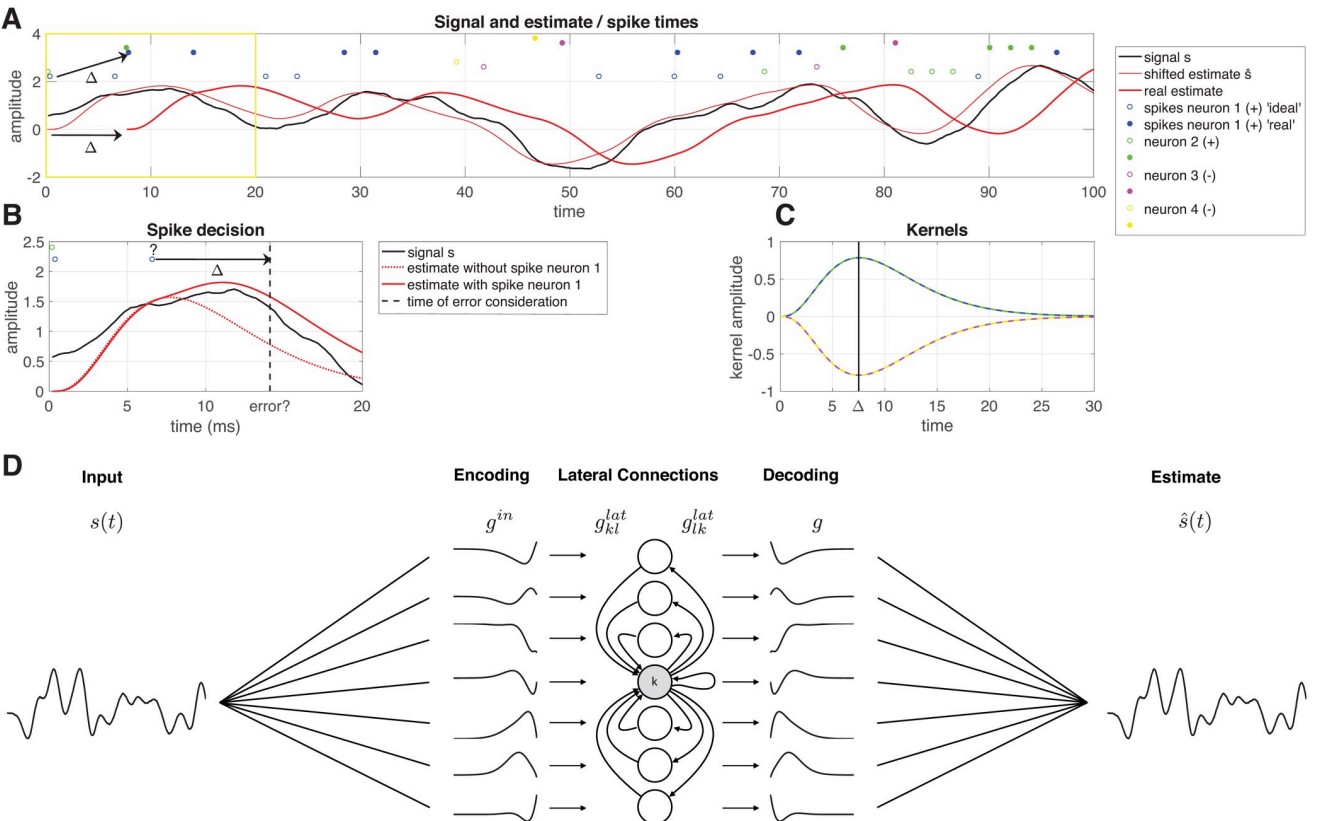

**Fig 1. Overview of the network model. A** The signal (black), ideal estimate $\hat{s}$ (thin red line), delayed estimate (thick red line), ideal spike trains $\rho^{\text{ideal}}$ (open dots) and real spike trains $\rho$ (filled dots). **B** Zoom in of the yellow square in A. The decision whether the blue spike (denoted with a question mark) in $\rho^{\text{ideal}}$ will be fired, is made at time $t + \Delta$. The spike is fired, because the estimate with the spike (solid red line) has a smaller error than the estimate without the spike (dashed red line). **C** Decoding filters $g$ used in this network: neurons 1 and 2 (blue and green) have positive filters, neurons 3 and 4 (pink and yellow) have negative filters. **D** Each neuron $k$ filters the input $s(t)$ using an input filter $g_k^{in}$. It receives input from other neurons $l$ via lateral filters $g_{kl}^{lat}$ and sends input to other neurons $l$ via lateral filters $g_{lk}^{lat}$. Finally, the network activity can be decoded linearly using decoding filters $g_k$.

We introduce a greedy spike rule: a spike will only be placed at time $T$ if this reduces the mean squared error at $T + \Delta$ (see Fig 1B), so if the expression in Eq (3) is positive. This results in a spike rule:

$$\int_0^{T+\Delta} g_m(t - T) \left( s(t) - \sum_{j=1}^{N} \sum_{i=1}^{n_j} g_j(t - t_j^i) \right) dt$$

$$> \frac{1}{2} \int_0^{T+\Delta} g_m(t - T)^2 dt. \tag{4}$$

Obviously, if the decision is made at time $t = T + \Delta$, a spike cannot be placed in the past ($t = T$). Therefore, we introduce two spike trains: the real spike train $\rho(t)$, where a spike will be placed at time $T + \Delta$ with the spike rule above (Eq (4)). This represents the 'ideal' spike train $\rho^{\text{ideal}}(t)$, which is $\rho(t)$ shifted by an amount of $\Delta$, so that a spike at time $T + \Delta$ in $\rho(t)$ is equivalent to a spike at time $T$ in spike train $\rho^{\text{ideal}}(t)$. This means that the neurons have to keep track of their

own spiking history for at least $\Delta$ time and that any prediction of the input is delayed by an amount of $\Delta$ (see Fig 1A, $\sum_{j=1}^{N} g_j(t) * \rho_j(t)$ is an estimate of the input, delayed $\Delta$ in time).

Eq (4) defines a filter-network (Fig 1D). Each neuron $m$ keeps track of a 'membrane potential' $V$

$$
\begin{aligned}
V_m(T + \Delta) = \quad & \int_0^{T+\Delta} g_m(t - T)s(t)dt \\
& - \int_0^{T+\Delta} g_m(t - T)\sum_{j=1}^{N}\sum_{i=1}^{n_j} g_j(t - t_j^i)dt,
\end{aligned}
\tag{5}
$$

that it compares to a threshold $\Theta_m$

$$
\Theta_m = \frac{1}{2}\int_0^{T+\Delta} g_m(t - T)^2 dt + v = \frac{1}{2}\int_{-T}^{\Delta} g_m(x)^2 dx.
\tag{6}
$$

Note that the threshold is not evaluated over the whole filter, but only between $t = -T$ and $t = \Delta$. Only if $\Delta$ is larger than the causal part of the filter and the proposed time of the spike $T$ larger than the acausal part, the whole filter is taken into account, and the threshold reduces to $\Theta_m = \frac{1}{2}$ for (L2-norm) normalized filters.

The network defined by Eq (4) can spike at any arbitrary frequency, and neurons show no spike-frequency adaptation. Two neurons that have identical filters except for their sign, have lateral filters $g_m^{\text{out}}(0) = 2(\Theta_m)$. Depending on the shape of the filter and the value of $\Delta$, the maximum value of these lateral filters can exceed twice the threshold of the postsynaptic neuron, so a spike in a neuron can induce a spike in a neuron with a filter that is identical except for their sign. This 'ping-pong effect', which makes the estimate fluctuate very quicky around the ideal value, can be dampened by introducing a spike cost. So to force the network to choose solutions with realistic firing rates for each neuron, we introduce two additional terms to the threshold:

$$
\Theta_m^c(t) = \frac{1}{2}\int_{-T}^{\Delta} g_m(x)^2 dx + v + \mu g^{\text{threshold}}(t_m - T),
\tag{7}
$$

where $v$ is a spike cost that punishes high firing rates in the network, which makes the code more sparse. $\mu g^{\text{threshold}}(t)$ is a spike cost kernel that punishes high firing rates in a single neuron. The effect is equivalent to spike-frequency adaptation [37]: every time a neuron fires a spike, it threshold is increased with $\mu$, and decays (with filter $g^{\text{threshold}}$) back to its original dynamic value. Compared to just adding a constant spike cost (as in citeBoerlin2011,Boerlin2013, Bourdoukan2012, Deneve2017), adding the temporal spike cost makes the network activity more distributed between the neurons: a neuron can fire a few spikes, but by doing that will increase its threshold, and another neuron will take over. In this paper we use an exponential kernel with time constant of 60 ms unless stated differently.

The first term on the right of Eq (5) shows that neuron $m$ is convolving the input $s(t)$ with an input filter $g_m^{\text{in}}(t)$ that is a flipped and shifted version of the filter neuron $m$ represents:

$$
\int_0^{T+\Delta} g_m(t - T)s(t)dt =
$$

$$
\int_0^{T+\Delta} g_m^{\text{in}}(T + \Delta - t)s(t)dt = (g_m^{\text{in}} * s)(T + \Delta),
$$

where

$$g_m^{\text{in}}(t) = g_m(\Delta - t).$$

(8)

Note that by introducing the time delay $\Delta$ between the evaluation time and the spike time, the input filter is now shifted relative to the representing filter. The representing filter can contain a causal part, that consists of the systems estimation on how the stimulus will behave in the near future of the spike, i.e. the systems estimation of the input auto-correlation. Any acausal part of the input filter is not used (since we do not know the future if the input $s(t)$), and vanishes as long as $g(t) = 0$ for $t > \Delta$. The optimal value of $\Delta$ depends on how much of a prediction the neuron wants to make into the future, but also on how long it is willing to wait with its response.

The second term of Eq (5) denotes the lateral $(j \neq m)$ and output $(j = m)$ filters, that are substracted from the filtered input as a result of the spiking activity of any of the neurons. For well-behaved filters we can write

$$\int_0^{T+\Delta} g_m(t - T) \sum_{j=1}^{N} \sum_{i=1}^{n_j} g_j(t - t_j^i) dt$$

$$= \sum_{j=1}^{N} \int_0^{T+\Delta} g_m(t - T)(g_j * \rho_j^{\text{ideal}})(t) dt$$

(9)

$$= \sum_{j=1}^{N} (g_m^{\text{in}} * g_j * \rho_j^{\text{ideal}})(T + \Delta),$$

so that we find an output filter

$$g_m^{\text{out}}(t) = -(g_m^{\text{in}} * g_m)(t)$$

(10)

and lateral filters

$$g_{mj}^{\text{lat}}(t) = -(g_m^{\text{in}} * g_j)(t),$$

(11)

where $t$ denotes the time since the spike of neuron $j$. Note that a spike at $T$ can only influence the decision process at $T + \Delta$, so the part of the filter where $0 \leq t - t_j^i < t_j^i + \Delta$ does not influence the spiking process, and is therefore not used.

In summary, we defined network that can perform near-optimal stimulus estimation (Fig 2). Given a set of readout filters $g_i(t)$, the membrane potential of each neuron $i$ is defined as

$$V_i(t) = g_i(t - \Delta) * \left( s(t) - \sum_{j=1}^{N} g_j(t) * \rho_j(t - \Delta) \right).$$

(12)

A spike is fired if this reduces the MSE of the estimate, which is equivalent to when the membrane potential exceeds the threshold $\Theta_m^c$ (Eq (7)). This model has two main characteristics: the input, output and lateral filters are defined by the representing filter, and the representation of the input by the output spike train is delayed by an amount $\Delta$ in order for the network to use predictions for the stimulus after the spike.

We conclude that a classical filter network can perform near-optimal stimulus encoding given a certain relation between the input, output and lateral filters. In the classical framework of LNP and GLM models, acausal filters $g$ are used, and the readout of the model is only used

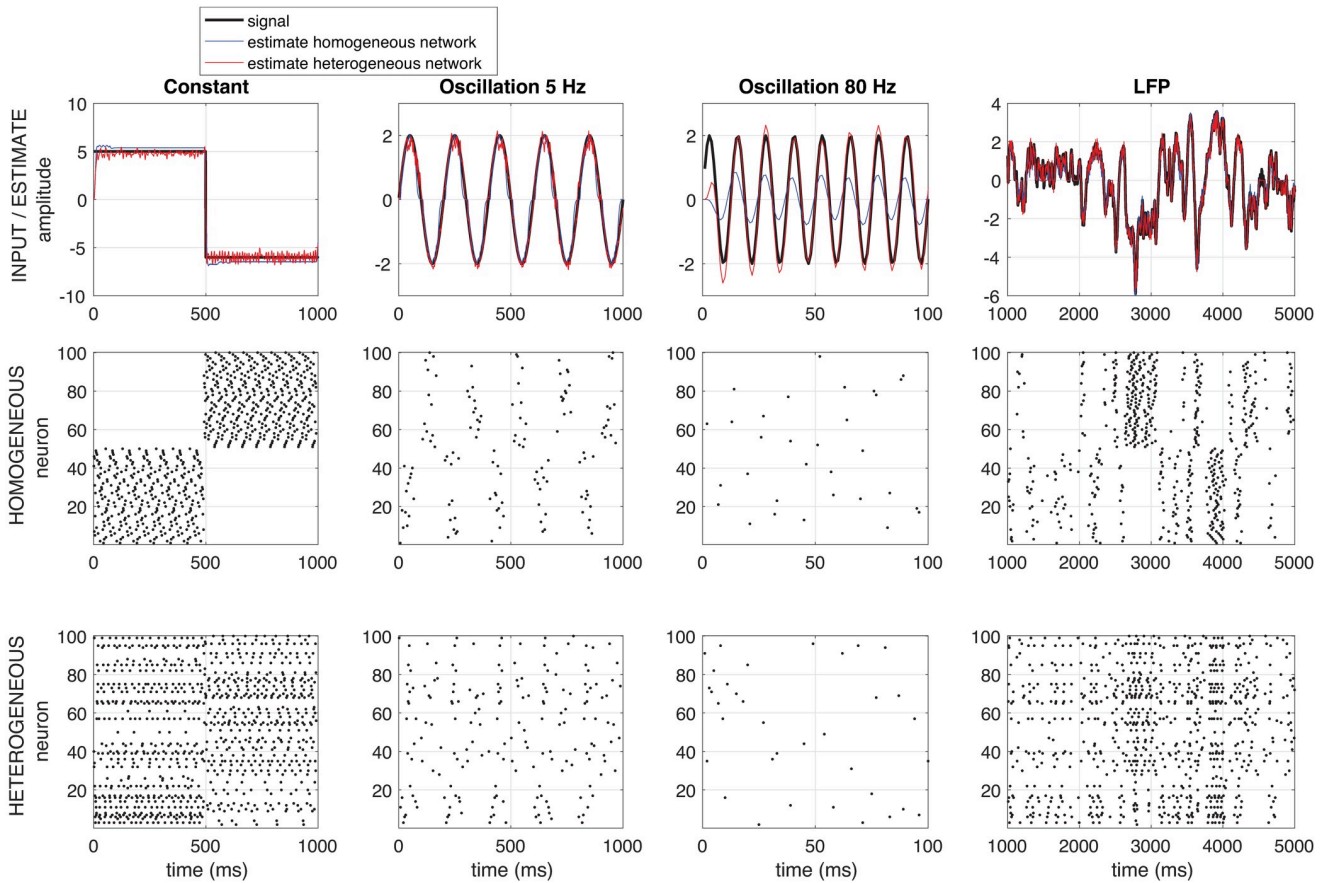

**Fig 2. Example network response.** Example of the response of a homogeneous (red) and a heterogeneous (blue) network to a constant input (first column), to a low frequency input (5 Hz, second colum), to a high frequency input (80 Hz, third column) and to a recorded LFP-signal (right). The homogeneous network consists of 50 neurons with a positive representing filter $g$ and 50 neurons with a negative one (see section 'Derivation of a filter-network that performs stimulus estimation); the heterogeneous network consists of 100 neurons with each a different (but normalized) representing filter $g$. Network parameters: $\Delta = 7$, 5 ms, $v = \mu = 0.5$. No noise is added to this network.

for information theoretic purposes. However, if the readout is being done by a next layer of neurons, or in recurrent networks, one cannot use acausal filters: every spike of a presynaptic neuron $m$ at time $t_m^i$, can only influence the membrane potential at its target at $t > t_m^i$ (i.e. can only cause a post-synaptic potential after the spike). Therefore, in a layered and recurrent networks, a self-consistent code will use only representative filters that are causal

$$g(t) = 0 \text{ if } t < 0$$

This means that for the input filters

$$g^{\text{in}} = 0 \text{ if } t > \Delta$$

Note that if $\Delta = 0$, the input filter reduces to a single value at $t = 0$. If $\Delta = 0$ and $g(0) = 0$, the input filter and hence the output and lateral filters vanish. Note that the threshold is scaled to the part of $g$ between 0 and $\Delta$, so not to the full integral over $g$. In this paper, we will normalize the input filters so that the thresholds are the same for all neurons.

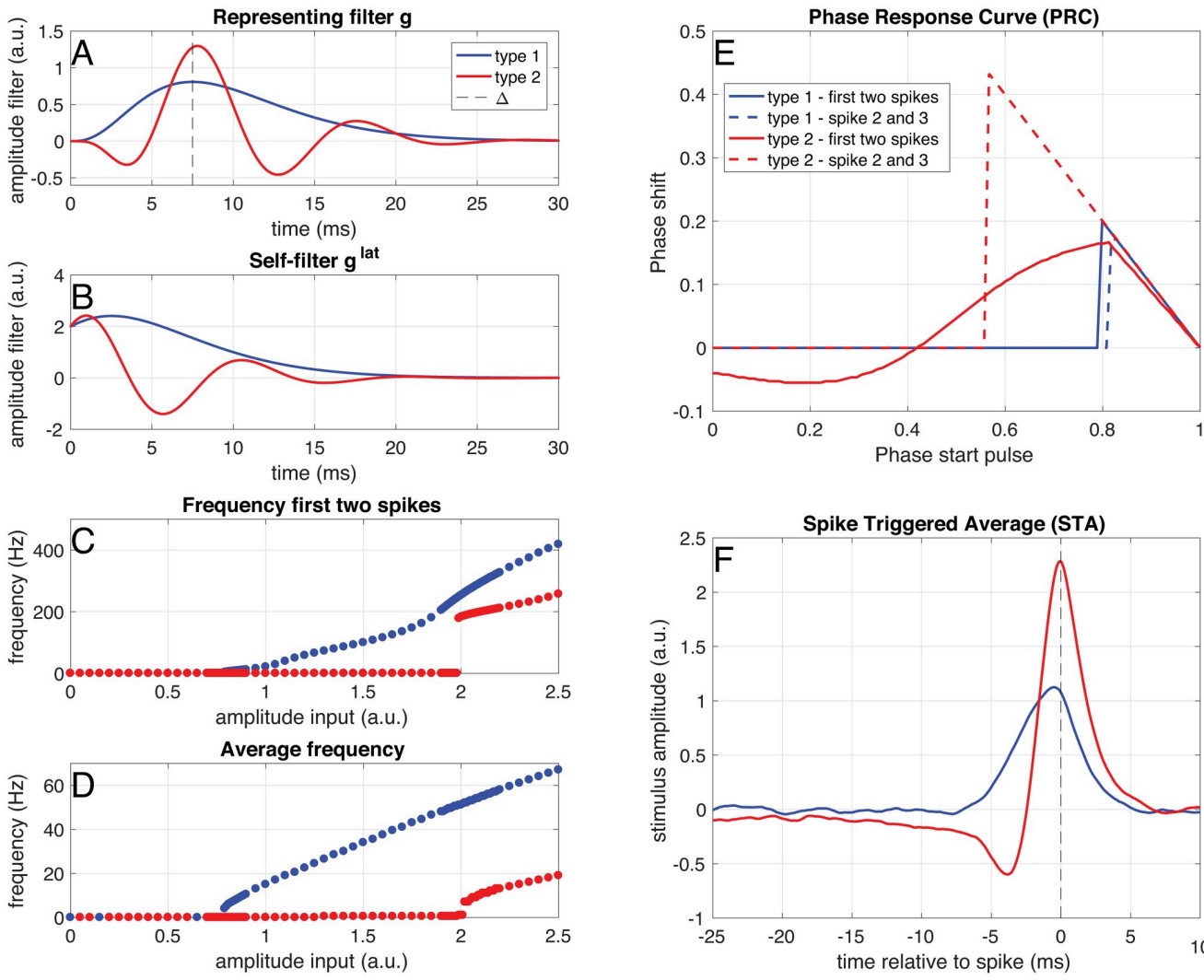

**Fig 3. Example of a 'type 1' (blue) and a 'type 2' (red) neuron. A** Representing filter of a 'type 1' (blue) and a 'type 2' (red) neuron. **B** Self-filters for both neuron types. **C** Instantaneous frequency of the first two spikes in response to step-and-hold inputs of different amplitudes. **D** Average frequency in response to step-and-hold inputs of different amplitudes. **E** Phase-response curves for the first and second pairs of spikes, calculated as response to a small pulse (0.1 ms, amplitude 1.5 ('type 2') and 3.7 ('type 2')) on top of a constant input (amplitude 0.9 ('type 1') and 2.2 ('type 2')). **F** Spike-triggered average in response to a white-noise stimulus (amplitude: 0.45 ('type 1') and 1.1 ('type 2')) filtered with an exponential filter with a time constant of 1 ms.

## Analysis

**Generation of neuron filters.**   In this paper, we choose each neuron filter on the basis of 8 basis functions given by the following Γ-functions:

$$\Gamma_n(t) = t^n e^{-t} \tag{13}$$

**Type 1 neuron** The representing filter of a 'type 1' neuron (Fig 3A and 3B, blue) is chosen equal to a Γ-function (Eq (13)) with $n = 3$.

**Type 2 neuron** The representing filter of a 'type 2' neuron (Fig 3A and 3B, red) is chosen as $g_{\text{type 2}}(t) = \Gamma_3(t)(0.2 - 0.8\sin(0.6t))$.

**Off cells** We call a neuron with an inverted representing filter an 'off cell'. So a 'type 1 off cell' has representing filter $g_{\text{type 1,off}}(t) = -\Gamma_3(t)$ and a 'type 2 off cell' has representing filter $g_{\text{type 2,off}}(t) = -\Gamma_3(t)(0.2 - 0.8\sin(0.6t))$.

**Homogeneous network** The representing filters of a 'type 1'-network are equal to the 'type 1' neuron (half of the neurons) or minus the 'type 1' neuron (other half of the neurons).

**Type 1 & type 2 network** A quarter of the representing filters of the neurons in the network are equal to the 'type 1' neuron, a quarter to minus the 'type 1' neuron, a quarter to the 'type 2' neuron and a quarter to minus the 'type 2' neuron.

**Heterogenous network** The representing filters in the heterogenous network are products of $\Gamma$ functions and oscillations. This is in order to ensure that they have realistic properties (i.e. vanishing after tens of miliseconds), while still being able to represent different frequencies. Half of the neurons have a representing filter equal to $g_n(t) = \Gamma_3(t)(0.2\pm0.8\sin(\psi t))$, with $\psi$ a random number between 0 and 1.5 and half of these using an addition and half a subtraction. The representing filters of the other half of the population are given by $g_n(t) = \Gamma_3(t)(0.2\pm0.8\cos(\psi t))$.

**Spike coincidence factor.** The coincidence factor $\Gamma$ [38, 39] describes how similar two spike trains $s_1(t)$ and $s_2(t)$ are: it reaches the value 1 for identical spike trains, vanishes for Poissonian spike trains and negative values hint at anti-correlations. It is based on the binning of the spike train in $K = \frac{T}{p}$ bins of binwidth $p$. The coincidence factor is corrected for the expected amount of coincidences $\langle N_{\text{coinc}}\rangle$ of spike train $s_1$ with a Poissonian spike-train with the same rate $v_2$ as spike train $s_2$. It gives a measure of 1 for identical spike trains, 0 if all coincidences are accidental and negative values for anti-correlated spike trains. It is defined as

$$\Gamma_{12} = \frac{N_{\text{coinc}} - \langle N_{\text{coinc}}\rangle}{\frac{1}{2}(N_1 + N_2)} \frac{1}{\mathbb{N}} \tag{14}$$

in which

$$\langle N_{\text{coinc}}\rangle = 2v_2 p N_1 = \frac{2N_1 N_2}{K}$$

Finally, $\Gamma$ is normalized by

$$\mathbb{N} = 1 - 2v_2 p = 1 - \frac{N_2}{K}$$

so it is bounded by 1. Note that the coincidence factor is not symmetric nor positive, therefore it is not a metric. It is only defined as long as each bin contains at most one event, however, we counted the bins with double spikes as bins containing one spike. Finally, it will in general saturate at a value below one, which can be seen as the spike reliability. The rate as which it reaches this value (for instance as defined by a fit to an exponential function) can be seen as the precision. In Section Heterogeneous networks are more efficient than homogeneous networks we calculated the coincidence factor between the spike-train response to each stimulus presentation for each neuron in the network, and averaged this over neurons to obtain a $\bar{\Gamma}$, a

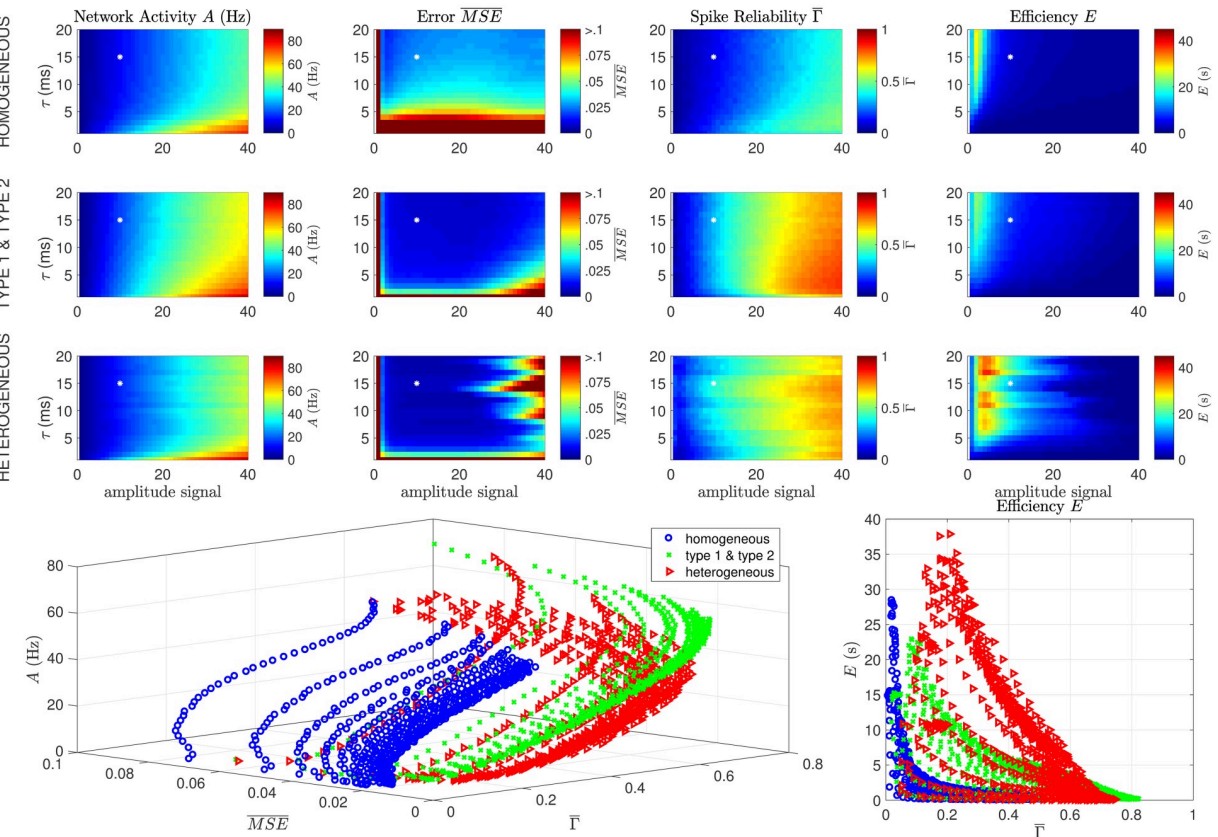

**Fig 4. eterogeneous networks are more efficient and show less trial-to-trial variability.** Results of two simulations using the same stimulus, but different initial network states, in a homogeneous 'type 1' network (first row), a network with 'type 1' and 'type 2' neurons (second row) and a heterogeneous network (third row). Each data point was simulated ten times and averaged. The network network activity $A$ (first column), spike reliability $\overline{\Gamma}$ (second column), network $\overline{MSE}$ (third column) and efficiency $E$ (fourth column) are shown as a function of the time constant ($\tau$) and amplitude of the stimulus. All networks perform well over a wide range of stimulus amplitudes and frequencies, but the heterogeneous network is more efficient for a wider range of input parameters, in particular for fast-fluctuating low amplitue signals. The bottom row shows the same data presented differently: every data point is the average over 10 trials of a single value of the input stimulus amplitude and $\tau$. This shows how the network activity, error and efficiency depend on the spike reliability $\overline{\Gamma}$. Parameters: $\Delta = 7, 5$ ms, $v = \mu = 1, 5, N = 100$, trials = 10. The white star denotes the parameter values used in section Heterogeneous networks are more efficient than homogeneous networks.

measure for the trial-to-trial variability or the degeneracy of the code of the network (Fig 4, left).

**Mean-squared error, network activity and effciency.** The Mean-Squared Error (MSE) is for $N$ measurements in time is defined by

$$MSE(s, \hat{s}) = \sum_{n=1}^{N} (s_n - \hat{s}_n)^2$$

However, this typically increases with the stimulus amplitude and length. To assess the performace of networks independently of stimulus amplitude, we normalized the mean-squared error by deviding it by the mean-squared error between the stimulus and an estimate of a constant zero signal (or equivalently, the MSE between the stimulus and network estimate if the

network would be quiescent, $\mathrm{MSE_{no\ spikes}}$):

$$\overline{\mathrm{MSE}} = \frac{\mathrm{MSE}}{\mathrm{MSE_{no\ spikes}}}. \tag{15}$$

An $\overline{\mathrm{MSE}}$ close to zero means a good performance, whereas a value close to one means performance that is comparable to a network that doesn't show any activity. Given that the goal of this network is to give an approximation of the input signal with the lowest number of spikes as possible, we define the network activity $A$ (in Hz) as

$$A = \frac{\mathrm{spikes}}{N_{\mathrm{neuron}} T_{\mathrm{sim}}}, \tag{16}$$

where $N$ is the number of neurons, and $T$ the duration of the stimulus, and the efficiency $E$ (in seconds) of the network as

$$E = \frac{1}{\overline{\mathrm{MSE}} \cdot A}. \tag{17}$$

Alternative measures of efficiency were also calculated (see Figs A and B in Section C of the supplementary S1 Text), but did not significantly alter our conclusions.

## Results

In this section, we will discuss the properties of the network derived in section Derivation of a filter-network that performs stimulus estimation. We will start with the general network behaviour, and show that it can track several inputs. Next, we will show that this framework provides a functional interpretation of 'type 1' and 'type 2' neurons. In the following sections, we will zoom in on the relation between trial-to-trial variability and the degeneracy of the code used, and on the network's robustness to noise. Finally, we will make experimental predictions based on the network properties.

### Network response

In Fig 2, the response of two different networks is shown: a homogeneous network, consisting of 50 neurons with a positive representing filter $g$ (see section Analysis) and 50 neurons with a negative one and a heterogeneous network (bottom), consisting of 100 neurons with each a different (but normalized, based on $\Gamma$-fuctions) representing filter $g$. Both networks can track both constant and fluctuating inputs with different frequencies well. Note that even though there is no noise in the network, the network response is quite irregular, like in *in-vivo* recordings. Note also that the heterogeneous network is better at tracking fast fluctuations. How well the different types of networks respond to different types of input, what the response properties of the networks are, and what the influence of the type of filter $g$ is, will be investigated in the following sections.

### 'Type 1' and 'type 2' neurons

If we create representing filters randomly (see section Analysis), they generally fall into one of two types: unimodal ones (only postitive or only negative) or bimodal ones (both a positive and a negative part). In this section, we will investigate the difference between neurons using these two types of representing filters.

In Fig 3 the response of a single neuron with a unimodal (blue) or multimodal (red) representing filter (both are normalized with respect to the input filter $g_{\mathrm{in}}$, so between $t = 0$ ms and $t$

= $\Delta$ = 7, 5 ms) to different types of input is shown (see section Analysis). Note that these simulations are for single neurons, so there is no network present, like in *in-vitro* patch-clamp experiments. A neuron with a unimodal representing filter (Fig 3A and 3B, blue) shows a continuous input-frequency curve (Fig 3C and 3D). Such a neuron with a unimodal representing filter has a unimodal Phase Response Curve (PRC) and Spike-Triggered Average (STA) (Fig 3E and 3F). A neuron with a multimodal representing filter (Fig 3A and 3B, red) does initially only respond with a single spike to the switching on of the step-and-hold current. Only for high current amplitudes it starts firing pairs of doublets, due to the interaction between the filtering properties and the spike-frequency adaptation. It has a bimodal PRC within the doublets (Fig 3E, solid red line), but a unimodal PRC between the doublets (Fig 3E, dashed red line). Such a neuron with a multimodal representing filter also has a bimodal Spike-Triggered Average (Fig 3F, red line). The input-frequency curves, PRC and STA together, show that neurons with unimodal representing filters show 'type-1' -like behaviour, whereas neurons with multimodal fitlers show 'type 2'-like behaviour.

## Homogeneous and heterogeneous networks

In the previous section, we showed that 'type 1' and 'type 2' neurons appear naturally in the predictive coding framework we defined in section Derivation of a filter-network that performs stimulus estimation. Even though the single-neuron response properties of 'type 1' and 'type 2' neurons have been studied extensively, most simulated network consist of a single or a few homogeneous populations of leaky integrate-and-fire ('type 1') neurons. Here, we will investigate the effect of heterogeneity in the response properties of single neurons on the network properties and dynamics. We will compare a homogeneous network consisting of 'type 1'-neurons, an intermediate network consisting of 'type 1' and 'type 2'-neurons, and a heterogeneous network (see section Analysis).

**Heterogeneous networks are more efficient than homogeneous networks.** The trial-to-trial variability of network responses depends critically on both the network structure and on the input stimuli used. For instance, it has been shown that (sub)cortical responses to stimuli with naturalistic statistics are more reliable than responses to other stimuli [40–46]. This suggests that, depending on the cortical area and the input statistics, neural networks can use codes that are highly degenerate or non-degenerate. For clarification, we define here a degenerate code as a code in which the stimulus can be represented with a low error by several different population responses. Therefore, a degenerate code will show a high trial-to-trial variability, or a low spike reliability. We hypothesize that a network consisting of neurons that represent similar features of the common input signal (i.e. several neurons have the same representing filter *g*) will use more degenerate codes than networks consisting of neurons that represent different features of the input signal (i.e. every neuron has a different representing filter *g*). So we hypothesize that homogeneous networks will show a higher trial-to-trial variability (i.e. a lower spike reliability). Here, we investigate the relation between trial-to-trial variability, network performance, input statistics and the network heterogeneity. We do this by simulating the response of three different networks with increasing levels of heterogeneity (a network consisting only of identical 'type 1' neurons, a mixed network consisting of 'type 1' and 'type 2' neurons and a heterogeneous network in which each neuron is different, see section Analysis) to input stimuli with different statistical properties (varying the amplitude and the autocorrelation time constant $\tau$).

In Fig 4, we simulated three networks (see also section Analysis): a homogeneous network (first row), a mixed network (second row, consisting for 50% of 'type 1' neurons (positive and negative filters) and for 50% of 'type 2' neurons (positive and negative filters) and a

heterogeneous network (third row). We varied both the amplitude and the time constant of the input signal (by filtering the input forwards and backwards with an exponential filter). To determine the level of degeneracy of the code the network uses, we performed the following simulations: we computed the network response to the same stimulus ($T = 2500$ ms) twice, but before this stimulus started, we gave the network a 500 ms random start-stimulus. Note that there was no noise in the network except for the different start-stimuli. We calculated four network performance measures:

**Network Activity $A$ (Hz)** We assessed the total network activity (Fig 4, first column), as the average firing frequency per neuron (see section Analysis).

**Spike Reliability $\Gamma$** The coincidence factor $\Gamma$ [38, 39] (see section Analysis) describes how similar two spike trains are: it reaches a value of 1 for identical spike trains, vanishes for Poisson spike trains and negative values hint at anti-correlations. We calculated the coincidence factor between the spike-train response to each stimulus presentation for each neuron in the network, and averaged this over neurons to obtain $\bar{\Gamma}$, a measure for the trial-to-trial variability of the network (Fig 4, second column). If the network uses a highly degenerate code, the starting stimulus will put it in a different state just before the start of the stimulus used for comparison, and the trial-to-trial variability will be high (low $\bar{\Gamma}$). On the other hand, if the network uses a non-degenerate code, the starting stimulus will have no effect, and the trial-to-trial variability will be low (high $\bar{\Gamma}$). Therefore, $\bar{\Gamma}$ (Fig 4, left) represents the non-degeneracy of the code: a $\bar{\Gamma}$ close to zero corresponds to a high degeneracy and a high trial-to-trial variability, and a $\bar{\Gamma}$ close to one a low degeneracy and a low trial-to-trial variability.

**Error $\overline{MSE}$** To assess the performace of the network, we calculated the normalized mean-squared error ($\overline{MSE}$ (Fig 4, third column), see section Analysis), so that a value of $\overline{MSE}$ close to zero means a good network performance, and a value close to one means performance that is comparable to a network that doesn't show any activity.

**Efficiency $E$ (s)** Given that the goal of this network is to give an approximation of the input signal with the lowest number of spikes as possible, we define the network efficiency (in seconds) as the inverse of the product of firing rate and the error (see section Analysis), so that the efficiency decreases with both the network activity and the error ($\overline{MSE}$ (Fig 4, fourth column).

In Fig 4, it is shown that all three networks (homogeneous, type 1 & type 2 and heterogeneous, respective rows) perform well (small error, second column) over a wide range of stimulus amplitudes and frequencies. The performance of the network depends strongly on the heterogeneity of the network and the characteristics of the stimulus: heterogeneous networks show a smaller $\overline{MSE}$ (second column), especially for fast-fluctuating (small $\tau$) input. However, this comes at the cost of a higher activity $A$ (first column), in particular at larger stimulus amplitudes. If we summarize this by the efficiency $E$ (fourth column), we see that the heterogeneous network is more efficient, in particular for low amplitude and fast fluctuating stimuli. Alternative measures of efficiency were also calculated (see section C of the supplementary S1 Text), but did not significantly alter our conclusions.

The amplitude of the stimulus relative to the amplitudes of the neural filters and the amount of neurons is important. This can be understood using a simplified argument: to represent a high-amplitude stimulus requires all the neurons to fire at the same time, which makes the code highly non-degenerate: Since the filter amplitudes are between 1 and 2, a stimulus with an amplitude (standard deviation) of 20 would need an equivalent number of the $N = 100$

neurons (the ones with a positive filter) to spike at the same time to reach the amplitude of the peaks. This results in a reliable, non-degenerate code (high $\bar{\Gamma}$). Alternatively, when a low-amplitude stimulus is represented by many neurons with relatively high-amplitude filters, the network can 'choose' which neuron to use for representing the input, thereby making the code degenerate (low $\bar{\Gamma}$). This argument does not take into account the temporal aspect of the representing filters (i.e. that each spike contributes an estimate into the future), the fact that representing filters can have positive and negative parts and the implemented spike-frequency adaptation, but the argument does show that the stimulus amplitude needs to be considerably lower than the sum of the absolute value of all the representing filters in the network.

Note that none of the networks is very good at responding to very low stimulus amplitudes: this is because most fluctuations are smaller than the filter amplitudes. This is reflected in the spike reliability $\bar{\Gamma}$ (third column). In a homogeneous network consisting of identical neurons representing a low amplitude input, there is no difference between a spike of neuron A or one of neuron B, making the code highly degenerate (low $\bar{\Gamma}$). When the amplitude of the stimulus increases, more neurons are recruited to represent the stimulus, thereby increasing both the network activity $A$ and the spike reliability $\bar{\Gamma}$. When the stimulus amplitude becomes too high to represent properly, both the error $\overline{\text{MSE}}$ and $\bar{\Gamma}$ increase sharply (bottom left), and the efficiency $E$ decreases (bottom right). Therefore, in our framework, a high trial-to-trial variability, or a low spike reliability $\bar{\Gamma}$ is a hallmark of an efficiently coding network, and a strong decrease of the trial-to-trial variability (an increase in $\bar{\Gamma}$) is a sign of a network starting to fail to track the stimulus. This happens for lower amplitudes for the heterogeneous network than for the homogeneous network.

In conclusion, all three networks can track stimuli with a wide range of parameters well, but the heterogeneous network performs shows a lower error for an only small increase in activity, therefore the heterogeneous network is more efficient than the homogeneous network. However, the homogeneous network can track higher amplitude stimuli, in particular slowly fluctuating ones. The efficiency of the code is high for intermediate values for the trial-to-trial variability: a very low trial-to-trial veriability (high $\bar{\Gamma}$) corresponds with a low efficiency, but so does a very high trial-to-trial variability ($\bar{\Gamma}$ close to 0). So in our framework, an high to intermediate trial-to-trial variability is a hallmark of an efficiently coding network.

**Heterogeneous networks are more robust against correlated noise than homogeneous networks.** In the previous section it was shown that heterogeneous networks are more efficient than homogeneous networks in encoding a wide variety of stimuli. However, these networks did not contain any noise. It has been shown that cortical networks receive quite noisy input, which is believed to be correlated between neurons [47–49]. Therefore, we will test in this section how robust homogeneous and heterogeneous networks are against correlated noise.

To test for robustness against noise, we chose a stimulus that all networks responded well to: amplitude = 10, $\tau$ = 15 ms (see the white start in Fig 4). Noise was added in a 'worse case scenario': it had the same temporal properties as the stimulus (i.e. it was white noise filtered with the same filter with $\tau$ = 15 ms) and several neurons received the same noise. Two parameters were varied: the amplitude of the noise, and the correlations between the noise signals that different neurons receive. This was implemented as follows: next to the stimulus, every neuron received a noise signal and we varied the relative amplitude of the noise signal ($a_{\text{noise}}/a_{signal}$). Correlations between the noise signals for different neurons were simulated by only making a limited amount of noise copies, that were distributed among the neurons. So in Fig 5, the top horizontal row of each axis corresponds to the situation in which each neuron receives an

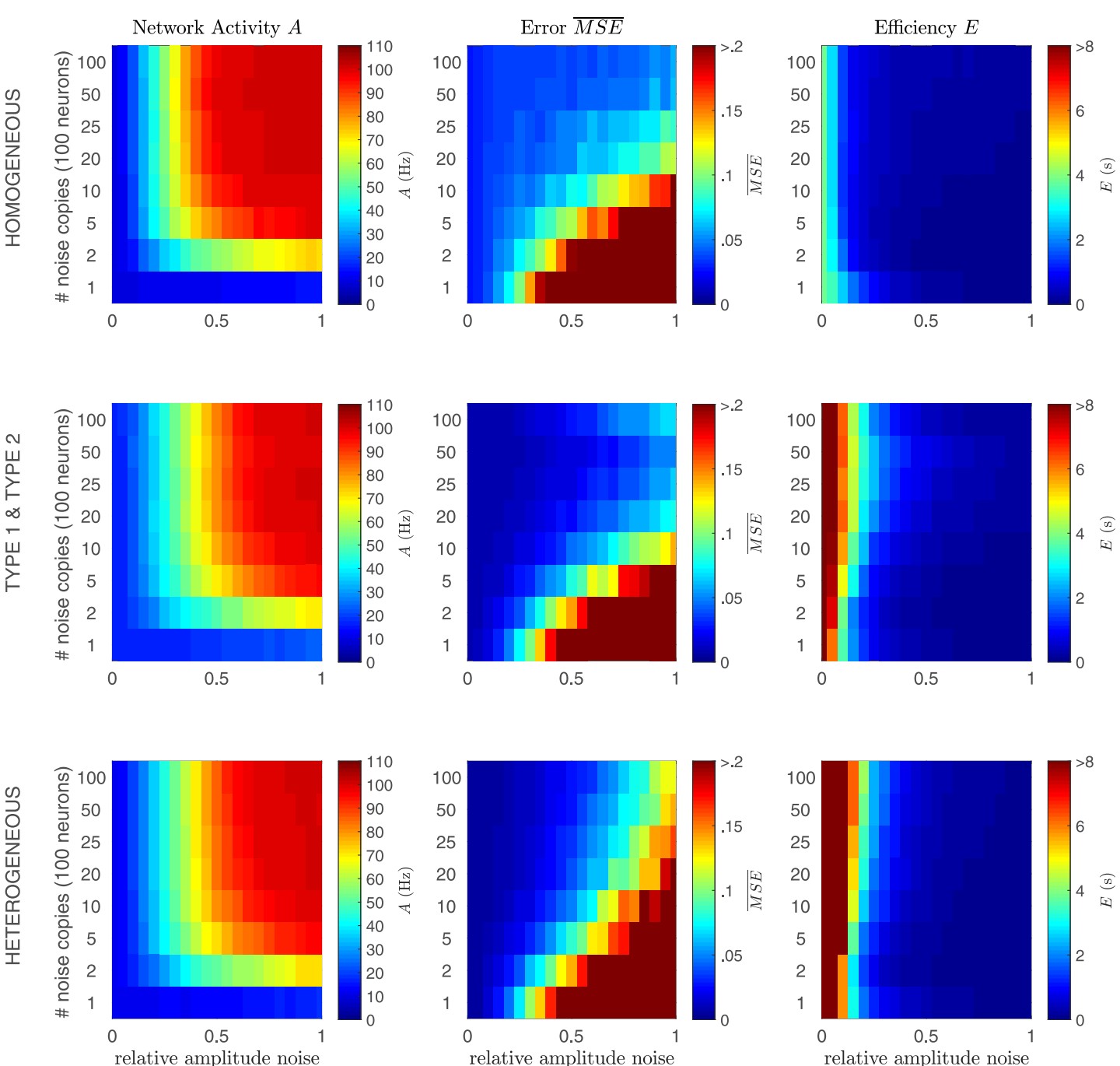

**Fig 5. Heterogeneous networks are more robust against noise.** Network activity $A$ (first column), network error $\overline{MSE}$ (second column) and efficiency $E$ (third column) of a homogeneous network (top row), a network with 'type 1' and 'type 2' neurons (middle row) and a heterogeneous network (bottom row). Next to the stimulus, each neuron in the network was presented with a noise input, with a varying relative amplitude (horizontal axis) and number of copies of the noise signal (vertical axis; 1 copy means all neurons receive the same noise, 100 copies means all neurons receive independent noise). Network: $\Delta = 7, 5$ ms, $\nu = \mu = 1, 5, N = 100$. Stimulus: amplitude = 10, $\tau = 15$ ms. Noise: $\tau = 15$ ms.

independent noise signal, the bottom horizontal row of each axis represents the situation where all neurons receive the same noise signal (and hence there is from the network's point of view no difference between signal and noise) and the leftmost column of each axis corresponds to the situation without noise.

In Fig 5, it is shown that all three networks are very effective in compensating for independent noise by increasing their firing rate. Note that the stimulus and the noise add quadratically, so that a signal with amplitude 10 and noise with amplitude 12 (relative amplitude 1.2) add together to a total input of amplitude $\sqrt{10^2 + 12^2} = 15.5$ for each neuron. However, the firing rate of a network with signal amplitude = 10 and noise amplitude = 12, is much higher than the firing rate of a network that receives a signal with amplitude 15.5 and no noise (compare Figs 4 and 5). So the networks increase their activity both due to the increased amplitude of the input, and in order to compensate for noise. The bottom row of each subplot in Fig 5 corresponds to a simulation where all neurons in the network recieve the same noise signal. In this case, it is impossible for the network to distinguish between signal and noise. In the row above, only two noise copies are present, in the row above that five, and so on. The homogeneous network can handle higher amplitudes of independent noise (top part of each subplot) before the representation breaks down ($\overline{\mathrm{MSE}} > .2$), but all networks are able to compensate for independent noise amplitudes up to equal to the signal amplitude (signal-to-noise ratio = 1). The heterogeneous network however, is better at dealing with correlated noise (bottom part of each subplot): it shows a lower error and higher efficiency for when there are few copies of the noise signal (alternative measures of efficiency were also calculated (see Fig A in the supplementary S1 Text), but did not significantly alter our conclusions). The type 1'& type 2 network appears to combine the properties of both networks: it has a low error in both representing independent noise and at representing correlated noise. These differences in robustness against noise probably have a relation with ambiguities in degenerate codes, as will be discussed in the Conclusion and discussion.

## Predictions for experimental measurements

In the previous sections, we derived a predictive coding framework and assessed the efficiency and robustness of representing a stimulus of homogeneous and heterogeneous networks. In this section, we will simulate often-used experimental paradigms, to assess what can be expected from such measurements with respect to the effect of the predictive coding framework on correlation structures.

**Signal and noise correlations.** The predictive coding framework that we derived in section Derivation of a filter-network that performs stimulus estimation predicts a specific correlation structure: neurons with similar filters $g$ should show positive signal correlations (because they use similar input filters), but negative noise (also termed spike count) correlations (because they have negative lateral connections). In many experimental papers, noise and signal correlations between neurons are measured [50, 51] (for an overview, see [52]). However, the methods authors use vary strongly: the amount of repetitions of the stimulus varies from tens to hundreds, the windows over which correlations are summed vary from tens of miliseconds to hundreds of miliseconds and the strength of the stimulus (i.e. network response) varies from a few to tens of Hz. All these parameters strongly influence the conclusions one can draw about correlations between (cortical) neurons. In order to be able to compare the correlations this framework predicts with experiments, we performed the following simulation (based on [51], Fig 6): we chose a 20 s. stimulus (exponentially filtered noise, $\tau = 15$ ms), and showed a network of $N = 100$ neurons 300 repetitions. We chose the stimulus amplitude so that the network response was around 8 Hz. In order to be able to compare neurons with similar filters and neurons with different filters, we used the 'type 1 & type 2' network (see section Analysis). To assess the effects of shared noise, each neuron received a noise signal that was the sum of an independent noise signal, and a noise signal that was shared between 10 neurons (amplitude signal = 2.7, amplitude independent noise = 0.5, amplitude correlated

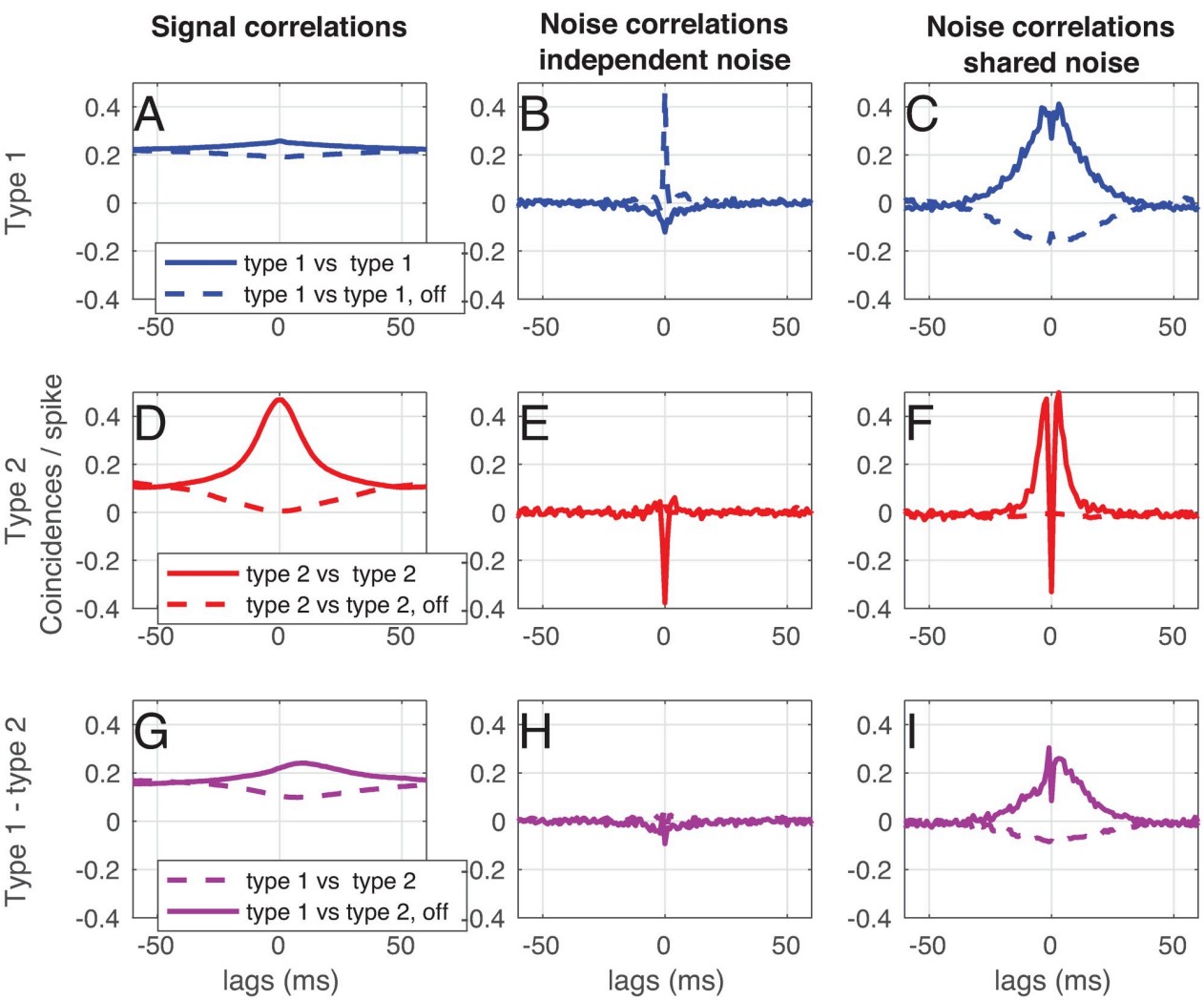

**Fig 6. Signal and noise correlations in a network with 'type 1' and 'type 2' neurons.** (see section Analysis and Fig 3). Next to the stimulus, each neuron in the network was presented with a noise input (of which half the power was independent, and half was shared with a subset of other neurons). A) Signal correlations between 'type 1' on-cells (solid blue line) and between a 'type 1' on and off cell (dashed blue line). B) Noise correlations between 'type 1' on-cells (solid blue line) and between a 'type 1' on and off cell (dashed blue line) receiving independent noise. C) Noise correlations between 'type 1' on-cells (solid blue line) and between a 'type 1' on and off cell (dashed blue line) receiving shared noise. D) Signal correlations between 'type 2' on-cells (solid red line) and between a 'type 2' on and off cell (dashed red line). E) Noise correlations between 'type 2' on-cells (solid blue line) and between a 'type 2' on and off cell (dashed red line) receiving independent noise. F) Noise correlations between 'type 2' on-cells (solid red line) and between a 'type 2' on and off cell (dashed red line) receiving shared noise. D) Signal correlations between a 'type 1' and a 'type 2' on-cell (solid purple line) and between a 'type 1' on and a 'type 2' off cell (dashed purple line). E) Noise correlations between a 'type 1' on-cell and a 'type 2' on-cell (solid purple line) and between a 'type 1' on and a 'type 2' off cell (dashed purple line) receiving independent noise. F) Noise correlations between a 'type 1' on-cell and a 'type 2' on-cell (solid purple line) and between a 'type 1' on and a 'type 2' off cell (dashed purple line) receiving shared noise. Network parameters: $\Delta = 7, 5$ ms, $v = \mu = 1, 5, N = 100, N_{\text{trial}} = 300$. Stimulus: $\tau = 15$ ms, amplitude = 2.7. Noise: $\tau = 15$ ms, amplitude independent noise = 0.5, amplitude shared noise = 0.5.

noise = 0.5). In this simulation, we can compare neurons that share a noise source, and neurons that don't.

In Fig 6, the signal and noise correlations between 'type 1' and 'type 2' and 'on' and 'off' cells (see section Analysis) and Fig 3) are shown. We ran a simulation consisting of 300 trials, in which the same signal, but a different noise realization was used. We used the same method as [49] (Note that the spike-count correlation is proportional to the area under the noise cross-correlogram in this method):

- For the **signal correlations** (Fig 6, first column), we calculated the average spike train over the 300 trials and calculated the cross-correllogram, normalized to the total average number of spikes.

- For the **noise correlations** (Fig 6, second column), we subtracted the cross-correlogram of the averaged spike trains (see above) from the cross-correlograms averaged over all trials.

- For the **noise correlations, shared noise** condition (Fig 6, third column), we simply calculated the noise correlations as above for two neurons that shared a noise source.

We performed this correlation analysis for two 'type 1' neurons (Fig 6, top row, blue), for two 'type 2' neurons (Fig 6, middle row, red) and for a 'type 1' and a 'type 2' neuron (Fig 6, bottom row, purple). We performed the correlation analysis also between two neurons with the same representing filter (solid line) and with a neuron and a neuron with an inverted representing filters ('off-cells', dashed lines).

We start by looking at cells with similar filtering properties. In Fig 6A, we show that 'type 1' neurons show positive signal correlations with other 'type 1' neurons (solid blue line), and negative signal correlations with their off-cells (dashed blue line). As expected, 'type 1'-neurons show negative noise correlations with other 'type 1'-neurons (Fig 6B, solid blue line), and positive noise correlations with their off-cells as long as noise is independent (dashed blue line). When neurons share a noise source (Fig 6C), the noise correlations are positive, but show a small negative deflection around zero lag for on cells (solid blue line). For 'type 2' neurons, we can show similar conclusions: two on-cells have positive signal correlations (Fig 6D, solid red line), negative noise correlations for independent noise ((Fig 6E, solid red line), and positive noise correlations with a negative peak around zero for shared noise ((Fig 6F, solid red line). An on and an off cell show negative signal correlations (Fig 6D, dashed red line), but hardly any noise correlations for independent noise (Fig 6E, dashed red line) or shared noise (Fig 6F, dashed red line).

We now focus at cells with different filtering properties. 'Type 1' and 'type 2' on cells show positive signal correlations (Fig 6G), solid purple line). Note that the peak is shifted towards positive lags, meaning that the 'type 2' neurons spike earlier in time than 'type 1' neurons. 'Type 1' and 'type 2' on cells show only small noise correlations for independent noise (Fig 6H, solid purple line). When noise is shared (Fig 6I, solid purple line), a small deflection at zero lag can be seen.

Under experimental conditions, correlations are often summed over a window of about 10 milliseconds or more. Therefore, we conclude that even though the predictive coding framework predicts negative noise correlations between similarly tuned neurons, and positive noise correlations between oppositely tuned neurons, these would be very hard to observe experimentally. For shared noise, we expect to see noise correlations with a similar sign as the signal correlations, but with small deflections at small lags, as shown in Fig 6C, 6F and 6I.

**'Type 2' neurons show more coherence with network activity.**   To examine how the two different neuron types couple to the network activity in response to a temporally fluctuating stimulus, we study the spike coherence with a simulated Local Field Potential (LFP). In Fig 7, we analyse the network activity of the different types of neurons, using the 'type 1 & type 2'-network (see section Analysis and Fig 3). Next to the stimulus, each neuron in the network was presented with a noise input (of which half the power was independent, and half was shared with a subset of other neurons). Network: $\Delta$ = 7, 5 ms, $\nu$ = $\mu$ = 1, 5, $N$ = 100. Stimulus: $\tau$ = 15 ms, amplitude = 2.7. Noise: $\tau$ = 15 ms, amplitude independent noise = 0.5, amplitude shared noise = 0.5. For the network activity, each spike was convolved with a Gaussian kernel ($\sigma$ = 6 ms). Looking at the network activity (Fig 7A and 7B), it is clear that 'type 2' neurons (red line) respond in a much more synchronized manner (high peaks), and 'type 1' neurons

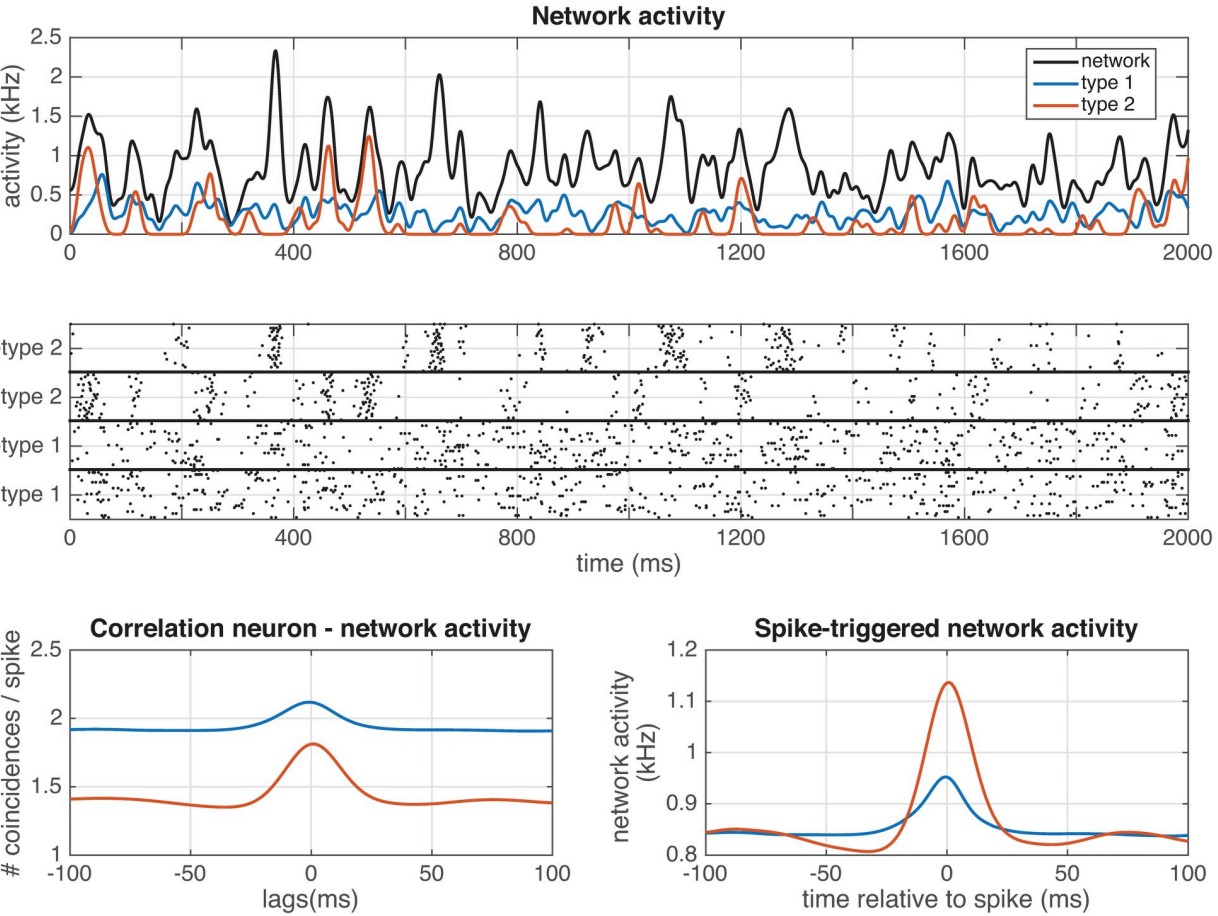

**Fig 7. Network activity in a 'type 1 & type 2' network.** (see section Analysis and Fig 3). A) Average activity of the whole network (black), 'type 1 on cells' (blue line) and 'type 2 on cells' (red line). B) Spike response of the network. C) Average cross-correllogram between the network activity and 'type 1, on' cells (blue line) or 'type 2, on' cells (red line). D) Spike-triggered network activity for 'type 1, on' cells (blue line) or 'type 2, on' cells (red line). Next to the stimulus, each neuron in the network was presented with a noise input (of which half the power was independent, and half was shared with a subset of other neurons). Network: $\Delta = 7$, 5 ms, $v = \mu = 1$, 5, $N = 100$. Stimulus: $\tau = 15$ ms, amplitude = 2.7. Noise: $\tau = 15$ ms, amplitude independent noise = 0.5, amplitude shared noise = 0.5. For the network activity, each spike was convolved with a Gaussian kernel ($\sigma = 6$ ms).

(blue line) respond in a much more continuous way. We quantified this, by convolving each spike with a Gaussian kernel ($\sigma = 6$ ms), and calculating both the correlations between the network activity and that of a single neuron (Fig 7C), and by calculating the 'spike-triggered network activity' [53] (Fig 7D). From this it is clear, that 'type 2' neurons show a much stronger coupling to the overall network activity (making them 'chorists') than 'type 1' neurons (making them 'soloists'). So we predict that 'type 2' neurons are more coherent with the network activity/LFP than 'type 1' neurons.

## Conclusion and discussion

Biological data often show a strong heterogeneity: neural properties vary considerably from neuron to neuron, even in neurons from the same network [1–5], but also see [54]. Theoretical networks, however, often use a single or only a very limited amount of 'cell types', where neurons from the same cell type have the same response properties. In order to investigate the

effects of network heterogeneity on neural coding, we derive a filter network that efficiently represents its input from first principles. We start with the decoding instead of with the encoding (this is not common, but has been done before [55]), and formulate a spike rule in which a neuron only fires a spike if this reduces the mean-squared error between the received input and a prediction of the input based on the output spike trains of the network, implementing a form of Lewicki's 'matching pursuit' [56]. Linear decoding requires recurrent connectivity, as neurons representing different features in the input should inhibit one another to alllow linear decoding, something that has been shown in experiments [57]. A similar framework has been formulated in homogeneous networks with integrate-and-fire neurons, the so-called 'Spike-Coding Networks'(SCNs) [58–61] and in networks using conductance-based models [62]. Effectively, this network performs a form of coordinate transformation [63]: each neuron represents a particular feature of the input, and only by combining these features the complete stimulus can be reconstructed. This network is related to autoencoders, in that it finds a sparse distributed representation of a stimulus by using an over-complete set of basis functions in the form of a feed-forward neural-network. The homogeneous integrate-and-fire networks in this framework have been shown to operate in a tightly balanced excitatory-inhibitory regime, where a large trial-to-trial variability coexists with a maximally efficient code [64]. Because of their linear read-out and filtering, only linear computations can be performed in this framework, which leaves open the questions whether the results presented here also hold for non-linear network computations [65, 66]. Recently, the SCN framework has been extended to include non-linear computations [67]. It is an interesting question to see whether the conclusions remain the same for non-linear computations.

With the derived filter network, we are able to study both single neuron and network properties. On the single neuron level, we find that the single-neuron response properties are equivalent to those of 'type 1' and 'type 2' neurons (for an overview see [8, 9, 68]): Neurons using unimodal representing filters showed the same behaviour as 'type 1' cells (continuous input-frequency curve [6], unimodal Phase-Response Curve (PRC) [7, 69] and a unimodal Spike-Triggered Average (STA) [70]), whereas neurons using bimodal filters correspond to 'type 2' cells (discontinuous input-frequency curve, bimodal PRC and a bimodal STA). This should be the case, as the STA is a result of the filtering properties of the neuron, and proportional to the derivative of the PRC [71]. In this framework, neurons with bimodal representing filters will also send bimodal Post-Synaptic Potentials (PSPs) to other neurons, so excitatory post-synaptic potentials (EPSPs) with an undershoot or inhibitory post-synaptic potentials (IPSPs) with a depolarizing part. This might sound counterintuitive, because we often think of EPSPs as having a purely depolarizing effect on the membrane potential of the postsynaptic neuron, and of IPSPs as having a purely hyperpolarizing effect. However, a post-synaptic potential might have both excitatory and inhibitory parts, depending on the type of synapse and the ion channels present in the membrane. For instance, an undershoot after an EPSP can be observed as an effect of slow potassium channels [17] such as $I_M$ [16], $I_A$ [15] or $I_{AHP}$ [72]. IPSPs can have direct depolarizing effects when the inhibition is shunting [73], [74], or due to for instance deinactivation of sodium channels, or slow activation of other depolarizing channels such as $I_h$. The observation that neurons using unimodal representing filters show 'type 1' behaviour and neurons using bimodal representing filters show 'type 1' behaviour gives a functional interpretation of these classical neuron types: 'type 1' cells are more efficient at representing slowly fluctuating inputs, whereas 'type 2' cells are made for representing transients and fast-fluctuating input. This can also be observed in the network activity: 'type 2' neurons show a much stronger coupling to the overall network activity (making them 'chorists' [53]) than 'type 1' neurons (making them 'soloists'). This is expected, as 'type 1' neurons are generally harder to entrain [69], whereas 'type 2' neurons generally show resonant properties [75]. So we predict

that 'type 2' neurons are more coherent with the network activity (local field potential) than 'type 1' neurons.

We compare both the functional coding properties and the activity of networks with different degrees of heterogeneity. We found that all networks in this framework can respond efficiently and robustly to a large variety of inputs (varying amplitude and fluctuation speed) corrupted with noise with different properties (ampitude, fluctuation speed, correlation between neurons). All networks show a high trial-to-trial variability, that decreases with the network efficiency. So we confirmed that in our framework trial-to-trial variability is not necessarily a result of noise, but is actually a hallmark of efficient coding [76–78]. In-vivo recordings typically show strong trial-to-trial variability between spike trains from the same neuron, and spike trains from individual neurons are quite irregular, appearing as if a Poisson process has generated them. In in-vitro recordings on the contrary, neurons show very regular responses to injected input current, especially if this current is fluctuating ([79, 80]. It is often argued that this is due to noise in the system and therefore that the relevant decoding parameter should be the firing rate over a certain time window (as opposed the timing of individual spikes, [81], but see also [82]). Here, we show in noiseless in-vivo-like simulations that the generated spike trains are irregular and show large trial-to-trial variability, even though the precise timing of each spike matters: shifting spike times decreases the performance of the network. Therefore, this model shows how the intuitively contradictory properties of trial-to-trial variability and coding with precise spike times can be combined in a single framework. Trial-to-trial variability is here a sign of degeneracy in the code: the relation between the network size, filter size and homogeneity of the network versus the amplitude determines whether there is strong or almost no trial-to-trial variability. Moreover, we show that the trial-to-trial variability and the coding efficiency depend on the frequency content of the input, as has been shown in several systems [40, 43, 45, 79].

In our framework, heterogeneous networks are more efficient than homogeneous networks, especially in representing fast-fluctuating stimuli: heterogeneous networks represent the input with a smaller error and using fewer spikes, in line with earlier research that found that heterogeneity increases the computational power of a network [23, 24], especially if they match the stimulus statistics [83]. Moreover, heterogeneous networks are not only more efficient, they are also more robust against correlated noise (noise that is shared between neurons) than homogeneous networks, in line with previous results [25, 84, 85]. This might be the result of heterogeneous networks using a less degenerate code: these networks are better at whitening the noise signal, because each neuron projects the noise onto a different filter, thereby effectively decorrelating the noise. Put differently, the heterogeneous network projects the signal and noise into a higher-dimensional space [63], thereby more effectively projecting noise and signal into different dimensions. Homogeneous networks are better at compensating for independent noise. This is probably due to a combination of two factors: 1) a homogeneous network being better at compensating for erronous spikes (a 'mistake' in the estimate is easier to compensate if the same but negative filter exist than if this doesn't exist) and 2) a homogeneous network is better at representing high-amplitude signals. The optimal mix of neuron types probably depends on the stimulus statistics and remains a topic for further study. The effects of neural diversity on computational properties of networks remains an open question (for a review, see [86]). It has for instance been shown that neural diversity can improve network information transmission [87] and results in efficient and robust encoding of stimuli by a population [4]. Recently, it has been shown that other forms of heterogeneity also result in advantages for network computation: diversity in synaptic or membrane time constants increases stability and robustness in learning [88] and diversity in network connectivity enables networks to track their input more rapidly [89], improves network stability and can lead to better

performance [90]. So diversity in network properties has shown to result in computational advantages on several levels and for different functions.

The predictive coding framework predicts a specific correlation structure between neurons: negative noise correlations between neurons with similar tuning, and positive noise correlations between neurons with opposite tuning. This may appear to be contradictory to earlier results [91, 92]. However, as neurons with similar tuning most likely recieve inputs from common sources in previous layers, these neurons will also share noise sources, resulting in correlated noise between neurons. We showed that in this situation, the negative noise correlations are only visible as small deflections of effectively positive correlations. It has been argued that in order to code efficiently and effectively, recurrent connectivity should depend on the statistical structure of the input to the network [93] and that noise correlations should be approximately proportional to the product of the derivatives of the tuning curves [94], although these authors also concluded that these correlations are difficult to measure experimentally. Others suggest that neurons might be actively decorrelated to overcome shared noise [95]. We conclude that even though different (optimal) coding frameworks make predictions about correlation structures between neurons, the opposite is not true: a single correlation structure can correspondd to different coding frameworks. So measuring correlations is not sufficient (although informative) to determine what coding framework is used by a network [35, 96, 97].

## Supporting information

**S1 Text. Section A**: **A word on notation**. **Section B**: **Technical notes**. **Section C**: **Efficiency measures**, **including Supplemental Figures A and B**. **Fig A**: **Comparison of efficiency measures** Results of two simulations using the same stimulus, but different initial network states, in a homogeneous 'type 1' network (first row), a network with 'type 1' and 'type 2' neurons (second row) and a heterogeneous network (third row). Three alternative efficiency measures are compared: 1) the network efficiency multiplied by the stimulus amplitude ($E_a$, see supp. eq. (10), first column), 2) the network efficiency multiplied by the stimulus power ($E_{a^2}$, see supp/ eq. (11), second column) and 3) the network cost ($C$, see supp. eq. (12), third column). The bottom row shows how the network efficiency or cost depends on the spike reliability $\bar{\Gamma}$. Parameters: $\Delta = 7, 5$ ms, $v = \mu = 1, 5$, $N = 100$, #trials = 10. The white star denotes the parameter values used in section 'Heterogeneous networks are more efficient than homogeneous networks' of the main text. **Fig B**: **Comparison of efficiency measures for the noise simulations** Efficiency normalized by amplitude $E_a$ (first column), amplitude squared $E_{a^2}$ (second column) and cost (third column) of a homogeneous network (top row), a network with 'type 1' and 'type 2' neurons (middle row) and a heterogeneous network (bottom row). Next to the stimulus, each neuron in the network was presented with a noise input, with a varying relative amplitude (horizontal axis) and number of copies of the noise signal (vertical axis; 1 copy means all neurons receive the same noise, 100 copies means all neurons receive independent noise). Network: $\Delta = 7, 5$ ms, $v = \mu = 1, 5$, $N = 100$. Stimulus: amplitude = 10, $\tau = 15$ ms. Noise: $\tau = 15$ ms. (PDF)

## Author Contributions

**Conceptualization:** Fleur Zeldenrust, Boris Gutkin, Sophie Denéve.

**Formal analysis:** Fleur Zeldenrust, Sophie Denéve.

**Funding acquisition:** Boris Gutkin, Sophie Denéve.

**Resources:** Fleur Zeldenrust.

**Supervision:** Boris Gutkin, Sophie Denéve.

**Writing – original draft:** Fleur Zeldenrust, Boris Gutkin, Sophie Denéve.

**Writing – review & editing:** Fleur Zeldenrust, Boris Gutkin, Sophie Denéve.

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
