## [Decision Letter · Decision Letter 0]

12 Feb 2021

Dear Dr Zeldenrust,

Thank you very much for submitting your manuscript "Efficient and robust coding in heterogeneous recurrent networks" for consideration at PLOS Computational Biology. As with all papers reviewed by the journal, your manuscript was reviewed by members of the editorial board and by several independent reviewers. The reviewers appreciated the attention to an important topic. Based on the reviews, we are likely to accept this manuscript for publication, providing that you modify the manuscript according to the review recommendations. In particular, by improving the description of the time-dependent threshold, clarifying the limitations of this approach, and - ideally - elaborating on the experimental predictions of the proposed framework.

Sincerely,

Daniel Bush

Associate Editor

PLOS Computational Biology

Daniele Marinazzo

Deputy Editor

PLOS Computational Biology

[LINK]

Reviewer's Responses to Questions

**Comments to the Authors:**

Reviewer #1: Overall comments:

This is a really nice paper that shows the advantages of heterogeneity for coding in a particular spiking neuron model. This is a pretty important point because the brain is heterogeneous but most models are not. I particularly liked the observation in the discussion that this is a noiseless in-vivo-like model in which trial-to-trial variability is high but the precise timing of each spike matters.

Major points:

- The argument rests on a model of neurons that produce spikes that perfectly minimise error when doing linear decoding. I think this is an interesting approach, but it might be nice to acknowledge limitations of that approach and discuss to what extent it can be implemented more practically.

- How robust are the results to the choice of efficiency measure? I can see the logic in dividing by number of spikes to get a measure of some sort of accuracy per spike, but 1/MSE as a measure of accuracy doesn't seem totally obvious to me. For example, you might equally argue that there is a cost to each inaccuracy, and a cost for each spike, and you want to minimise the sum of those costs, suggesting that you want to minimise a*num_spikes+b*MSE for some a, b. It would be nice to see that the conclusions about efficiency are robust to the measure chosen here, in the absence of any obvious standard measure.

- Conclusions are a bit too strongly worded, e.g. "high trial-to-trial variability is a hallmark for an efficiently coding network". In the specific case studied here, yes, but I'm not sure you can generalise this.

- On that specific point (on p8, end of second paragraph of right column), it looks like from Fig 4, bottom right, that actually there isn't quite an inverse relation of E with Gamma, but rather that it peaks around Gamma=0.2?

- Fig 4, heterogeneous MSE panel, what is going on at the right hand side of this figure? Why do some values of tau seem to have (randomly) much smaller or larger errors? Is this noise? Is it feasible to re-run the simulations more times to get a smoother figure?

Minor:

- On p8, left column, there's a combinatorial argument that you would expect the lowest reliability for smallest amplitude signals because the network can choose which neuron to use. But in the simple case of all amplitudes being the same, you would have the maximum number of choices for k where n choose k was largest, i.e. k=n/2. I suspect that I'm making a mistake here though.

- p8, bottom right, some sentences/parts of sentences repeated here.

- Section numbers are missing throughout.

Reviewer #2: ### Review summary

This work presents a conceptual novelty that combines heterogeneous neural temporal filters within an efficient coding framework. The analysis of the error and contribution of different properties of noise is interesting and shows unexpected results. The theory's predictive power and its use in future experiments is the weak spot of the work. One critical point that must be addressed is improving the description and implementation of the time-dependent threshold introduced. I recommend publishing this work after some revisions.

### Main contributions

1. This work's main conceptual novelty is considering the effect of using mixed temporal filters for predictive coding. Notably, some recent papers have studied related questions: In [1], the authors considered the benefits of using a population of neurons with heterogeneous time constants on learning. The study in [2] focused on the efficient predictive coding framework with random tuning (random weights) but a uniform temporal profile [2]. This work, for the first time to my knowledge, presents a model for efficient coding in spiking where a temporal filter defines neural tuning. I think this idea can influence future studies beyond the results of the current study.

2. The studies mentioned above demonstrated the benefits of heterogeneity: In [1], different temporal properties of neurons improve the learning of time-dependent data, and in [2], structural disorder improves the stability and can lead to better performances. The current study presents similar nature results, but I find them complementary and not overlapping: a network composed of neurons with different temporal filter can be more efficient.

3. The work provides an interesting analysis of different noise properties' contribution to the network's efficiency and correlation structure. First, they show that a heterogeneous network with mixed temporal profiles is more robust to correlated noise than the homogeneous counterpart. Second, the authors show that correlated noise –which is expected– hides the negative noise correlation predicted by the efficient coding network. The behavior of the model to different noise properties is beneficial for future studies.

### Criticism

1. The authors propose a solution to the known "ping-pong" effect of the efficient coding paradigm by adjusting the threshold. Increasing the threshold is a known solution and was proposed in the early works on efficient coding (ref [58] in the paper). In the current paper, the authors define a time-dependent threshold that acts as a filter. This solution is an interesting new view and seems plausible by considering an adaptation of neurons' firing rates. However, I find the explanation of the model lacking. In particular, it is not clear to me how the temporal kernel of the threshold is implemented. Is the kernel convoluted over the inputs or the membrane potential? The authors need to elaborate on how they implement the temporal-kernel threshold. Does it offer benefit when compared to the solution proposed by ref [58]? It is hard to judge the contribution and the biological plausibility without a clear description of the model.

2. The authors devote a large section to the experimental predictions of their theory. The current state of efficient coding theory lacks testable predictions, which will help advance and disseminate the ideas. Here, the authors describe in detail the results of numerical simulations that use biological values for the parameters. Considering it is a simplified model, I could not understand what the actual biological prediction is. The authors indicate that we should not expect to see the expected negative noise correlations because of correlated noise. While important, it is a postdiction of what we do not see in experiments, rather than a testable prediction. I find the experimental predictions the weakest part of this work.

### Minor issues

1. The authors note that in deep learning, causal filters are needed because of the layered structure and information flow direction. In my view, causal filters are also needed for a recurrent network. Since the authors are proposing a mechanistic model of neural coding –and not a statistical description– causality seems important for recurrent networks.

2. I find some of the definitions slightly confusing:

a. First, the authors define reliability as the similarity between spike trains. Naturally, one expects reliability to depend on the readout. In that case, a network that can produce the same readout with low error for very different spike trains is more reliable. Counterintuitively, the 'reliability' of the network is very prone to noise. I feel that 'consistency' is a more appropriate term. If the authors choose to keep the term 'reliability,' they may want to emphasize that difference.

b. The authors do not normalize the efficiency by the signal amplitude, so high-amplitude signals with high activity give low efficiency (which they notice in the results). It is counterintuitive to the meaning of efficiency, and I think it should be pointed out.

3. I could not understand the bottom-left panel of figure 4. Did the authors change the signal strength in different simulations to get different points? Is it just another way of presenting the top panels? I need a more explicit explanation of what is the take-home message from it.

### Recommendation

This work provides a conceptual novelty using temporal filters and tuning curves together is an efficient coding framework; this alone makes this work publication-worthy. The experimental predictions section is weak, and I would have liked to see some more clearly stated predictions, but I understand it is a lot to ask and would still recommend publication without it. There are some minor corrections to clarity and definitions that could be addressed. The one fault I believe must be addressed before publication is a better explanation of the time-dependent threshold and its implementation, and I recommend publishing the work after resubmission.

### Typos and errors

1. All the section references are missing.

2. The authors use the acronyms EPSP and IPSPS but define them only a few sentences after first use.

3. On page 9, the paragraph below the figure, third row. The reference to figure 5 points to the top row, where it should be the bottom row.

### References

[1] Perez-Nievez et al., Neural heterogeneity promotes robust learning. BioArxiv 2020, (https://www.biorxiv.org/content/10.1101/2020.12.18.423468v2.full)

[2] Kadmon et al., Predictive Coding in balance neural network with noise, chaos and delays, Neurips 2020 (https://papers.nips.cc/paper/2020/hash/c236337b043acf93c7df397fdb9082b3-Abstract.html)

**Have all data underlying the figures and results presented in the manuscript been provided?**

Reviewer #1: Yes

Reviewer #2: Yes

PLOS authors have the option to publish the peer review history of their article (what does this mean?). If published, this will include your full peer review and any attached files.

Reviewer #1: No

Reviewer #2: **Yes: **Jonathan Kadmon
---

## [Decision Letter · Decision Letter 1]

7 Apr 2021

Dear Dr Zeldenrust,

We are pleased to inform you that your manuscript 'Efficient and robust coding in heterogeneous recurrent networks' has been provisionally accepted for publication in PLOS Computational Biology.

Best regards,

Daniel Bush

Associate Editor

PLOS Computational Biology

Daniele Marinazzo

Deputy Editor

PLOS Computational Biology

Reviewer's Responses to Questions

**Comments to the Authors:**

Reviewer #1: The authors have fully responded to all my points and I'm happy with the paper as it is.

Congratulations on a great paper!

Reviewer #2: The authors have addressed my concerns, and I recommend publishing the paper in its current form.

**Have all data underlying the figures and results presented in the manuscript been provided?**

Reviewer #1: Yes

PLOS authors have the option to publish the peer review history of their article (what does this mean?). If published, this will include your full peer review and any attached files.

Reviewer #1: No

Reviewer #2: No

**Have the authors made all data and (if applicable) computational code underlying the findings in their manuscript fully available?**

Reviewer #2: Yes

---

## [Editor Report · Acceptance letter]

27 Apr 2021

PCOMPBIOL-D-21-00095R1 

Efficient and robust coding in heterogeneous recurrent networks

Dear Dr Zeldenrust,

I am pleased to inform you that your manuscript has been formally accepted for publication in PLOS Computational Biology. Your manuscript is now with our production department and you will be notified of the publication date in due course.

With kind regards,

Andrea Szabo
